# Vision-Language Models are Strong Noisy Label Detectors

**Tong Wei**[1,2,3]    **Hao-Tian Li**[1,2]    **Chun-Shu Li**[1,2]    **Jiang-Xin Shi**[3,4]
**Yu-Feng Li**[3,4]    **Min-Ling Zhang**[1,2]

[1]School of Computer Science and Engineering, Southeast University, Nanjing, China
[2]Key Laboratory of Computer Network and Information Integration (Southeast University),
Ministry of Education, China
[3]National Key Laboratory for Novel Software Technology, Nanjing University, China
[4]School of Artificial Intelligence, Nanjing University, China
{weit,liht}@seu.edu.cn

## Abstract

Recent research on fine-tuning vision-language models has demonstrated impressive performance in various downstream tasks. However, the challenge of obtaining accurately labeled data in real-world applications poses a significant obstacle during the fine-tuning process. To address this challenge, this paper presents a **De**noising **F**ine-**T**uning framework, called DEFT, for adapting vision-language models. DEFT utilizes the robust alignment of textual and visual features pre-trained on millions of auxiliary image-text pairs to sieve out noisy labels. The proposed framework establishes a noisy label detector by learning positive and negative textual prompts for each class. The positive prompt seeks to reveal distinctive features of the class, while the negative prompt serves as a *learnable threshold* for separating clean and noisy samples. We employ parameter-efficient fine-tuning for the adaptation of a pre-trained visual encoder to promote its alignment with the learned textual prompts. As a general framework, DEFT can seamlessly fine-tune many pre-trained models to downstream tasks by utilizing carefully selected clean samples. Experimental results on seven synthetic and real-world noisy datasets validate the effectiveness of DEFT in both noisy label detection and image classification tasks. Our source code is available at `https://github.com/HotanLee/DeFT`.

## 1 Introduction

Vision-language models pre-trained on large-scale image-text pairs, such as Contrastive Language-Image Pretraining (CLIP) [29], have gained widespread adoption in various machine learning tasks including few-shot learning [53, 52], multi-label learning [33, 14], and long-tail recognition [7, 30]. Recently, CLIP has shown impressive generalization performance to many downstream tasks without the need for adaptation [36, 31]. Zero-shot CLIP leverages the strong alignment of learned visual and textual features and classifies unseen images by comparing the similarity of image embedding with textual class prompts. Despite the good zero-shot performance, fine-tuning becomes necessary when the data distribution of downstream tasks significantly deviates from the CLIP training source. Two popular fine-tuning paradigms are commonly employed for adaptation, i.e., full fine-tuning (FFT), which modifies all model parameters, and parameter-efficient fine-tuning (PEFT), which fixes the pre-trained parameters while adding a few learnable parameters for adaptation. While promising, fine-tuning CLIP necessitates perfectly labeled datasets which may not be readily available in many real-world tasks, thereby hindering their broader applicability.

To tackle this issue, this paper investigates the fine-tuning of CLIP utilizing imperfectly labeled datasets, i.e., datasets with noisy labels. Intuitively, direct adaptation using noisy labels can signifi-

cantly degrade the performance. To mitigate the negative impact of noisy labels, researchers have proposed various approaches for learning with noisy labels, including sample selection techniques [10, 44, 43] and noise-robust learning methods [37, 27]. However, the exploration of this problem in the context of CLIP adaptation remains limited. This paper aims to fill this research gap by demonstrating that CLIP can be leveraged as effective noisy label detectors, capitalizing on their strong alignment between visual and textual features. To ensure the generality, we follow the basic idea in mainstream studies of learning with noisy labels. Firstly, we identify potentially mislabeled samples, and subsequently, we adapt the model parameters using carefully selected clean samples.

In the first phase, we propose to construct a noisy label detector by learning positive and negative textual prompts for each class. The positive prompt aims to uncover distinguishable features specific to the class, while the negative prompt functions as a learnable "threshold" for identifying noisy labels. Our noisy label detector can be derived from the similarity between embeddings of a training image and positive/negative prompts associated with its assigned class. If the image exhibits a higher similarity score with the positive prompt than the negative prompt, it is considered a correctly labeled sample, i.e., a clean sample. This design draws inspiration from previous studies that have shown the robustness of prompt tuning to noisy labels, particularly in the presence of high noise ratios [41]. To optimize the positive prompt, we align its representation with image features from the corresponding class during training. For the negative prompts, we introduce a novel negative learning objective. It is important to note that calculating similarities between images and textual prompts requires a strong alignment between these two modalities in the downstream task. To achieve this, we utilize PEFT, such as Visual Prompt Tuning (VPT) [18], for the adaptation of the visual encoder. This decision is based on a key finding that FFT can distort pre-trained feature representations when noisy labels are present, which is discussed in Section 3.2.

In the second phase, we can adapt the pre-trained model after acquiring carefully selected clean samples. It is worth noting that not only can we employ pre-trained CLIP for adaptation, but we can also utilize other pre-trained visual backbones such as ResNet [12] and MAE [11]. While the model adaptation phase may seem unnecessary since we can leverage the learned textual prompts for zero-shot classification without further fine-tuning, we emphasize its necessity by the improved performance on curated downstream datasets, particularly in fine-grained classification tasks. In this step, we learn a linear classifier and fully fine-tune the visual encoders using only selected clean samples. The rationale behind this approach is twofold: 1) the linear classifier tends to generalize better than textual class prompts, and 2) full fine-tuning can effectively adapt model parameters to address significant domain shifts, surpassing the capabilities of PEFT. In our implementation, the training is conducted for only 10 epochs, resulting in a minimal increase in computation cost.

Our main contributions can be summarized as follows:

- We propose DEFT, a simple yet effective framework for learning with noisy labels. DEFT offers several compelling advantages over existing methods: instance-dependent (no information required from entire training data), robust to various types of noisy labels, and generalizable to many pre-trained models.

- We conduct extensive experiments and show that DEFT achieves superior performance in both noisy label detection and image classification tasks on a wide range of synthetic and real-world datasets.

- We provide in-depth empirical analysis, providing insights to understand the effectiveness of DEFT. We hope that this work will serve as a springboard for future works on noisy label detection with multi-modal features.

## 2  Related Work

**Learning with Noisy Labels** A diverse variety of approaches have been proposed for addressing learning with noisy labels and can be broadly divided into twofold: robust learning and clean sample selection [9, 32]. Due to the susceptibility of traditional cross-entropy loss to noisy labels, several methods adopt noise-robust loss functions [8, 51, 37, 6] and regularization techniques [27, 38, 16] to enhance model robustness. For example, symmetric cross-entropy loss [37] introduces a noise-robust objective function for fast convergence and better performance. Early-learning regularization [27] counteracts the influence of the noisy labels by introducing a regularization term that integrates estimated target probabilities. However, empirical findings suggest that robust model learning faces

the problem of maintaining a balance between robustness and accuracy [51, 35]. Conversely, sample selection approaches strive to select clean samples from noisy datasets using specific criteria. Based on the memorization effect identified in deep neural networks [2], Numerous methods adopt the small-loss strategy that treats samples with small loss as clean ones [10, 15, 25, 19]. Co-teaching [10] selects a fraction of training data in each mini-batch based on the noise ratio, whereas DivideMix [25] fits a Gaussian mixture model to dynamically divide the training samples. Although these methods demonstrate strong performance, they inevitably overlook the significance of clean hard samples with a large loss. Besides, all the aforementioned approaches concentrate exclusively on learning from scratch with label noise. Our proposed approach leverages the pre-trained multi-modal information from visual-language models to construct a noisy label detector, resulting in improved sample selection performance.

**Fine-Tuning Vision-Language Models** Recent years have witnessed incredible progress in pre-trained visual-language models [29, 17, 49]. As a representative method, CLIP [29] employs contrastive pre-training to acquire informative representations from extensive image-text pairs. Despite the remarkable zero-shot performance of CLIP, its sensitivity to hand-crafted textual prompts is significant. Prompt-tuning aims to learn the optimal prompts from target data by replacing the textual templates with adaptable soft prompts, which demonstrates superior performance in the context of few-show learning [53, 52, 20] and multi-label learning [33, 14]. However, further enhancements can be achieved by adapting the visual encoder to accommodate larger datasets in downstream tasks. A standard approach to obtain task-specific representations is fine-tuning all model parameters. While FFT yields significant performance on curated datasets, recent work [1] empirically finds that FFT can result in feature distortion in the presence of massive noisy labels. In contrast, PEFT [48] has been verified to improve generalization capabilities for mitigation of overfitting issues [30], which fix the pre-trained model and introduce a few number of learnable parameters for adaptation, such as VPT [18], LoRA [13], and AdaptFormer [3]. Considering the distinct property of fine-tuning methods, this paper aims to delineate the most effective paradigm for model adaptation on noisy downstream datasets, which remains under-explored yet holds significant promise.

## 3 Preliminary and Initial Findings

### 3.1 Zero-Shot CLIP

This paper focuses on a $K$-class image classification task, where the objective is to categorize an image $\boldsymbol{x} \in \mathbb{R}^d$ and assign it a label $y \in [K] = \{1, 2, \ldots, K\}$. Benefiting from large-scale datasets and high-capacity models, CLIP [29] has demonstrated remarkable performance in zero-shot classification by computing the similarity between visual and textual representations. Specifically, with hand-crafted textual prompts such as "a photo of a [CLS]", where the class token is replaced by a specific class name, the prediction probability is computed as:

$$p(y = k \mid \boldsymbol{x}) = \frac{\exp(\texttt{sim}(\boldsymbol{I}, \boldsymbol{T}_k)/\tau)}{\sum_{k=1}^{K} \exp(\texttt{sim}(\boldsymbol{I}, \boldsymbol{T}_k)/\tau)}, \tag{1}$$

where $\texttt{sim}(\cdot, \cdot)$ denotes the cosine similarity between the extracted image features $\boldsymbol{I}$ and text features $\boldsymbol{T}$, and $\tau$ is the softmax temperature.

While zero-shot classification with CLIP is promising, its performance can be further enhanced by adapting CLIP on downstream labeled datasets $\mathcal{D} = \left\{(\boldsymbol{x}_i, y_i)_{i=1}^{N}\right\}$ to incorporate more task-specific representations. In this paper, we investigate the presence of noisy labels in the training dataset $\mathcal{D}$, which is prevalent in real-world applications. We use $r \in [0, 1]$ to denote the noise ratio of a training set, meaning that $N \times r$ training samples are incorrectly labeled, i.e., the assigned label $y$ differs from the ground-truth label $y^*$. Since $y^*$ is not accessible, our goal is to fine-tune the CLIP model using training labels $y$ by eliminating the negative influence of noisy labels. Considering the feature distortion induced by noisy labels [2], the key challenge lies in obtaining distinguishable yet robust representations when adapting CLIP on noisy datasets.

### 3.2 Fine-tuning CLIP on Downstream Datasets

To discern the most effective method for CLIP adaptation, we conduct empirical studies on various datasets as shown in Figure 1. Specifically, we utilize three fine-tuning approaches to adapt the

pre-trained CLIP model and compare their performance on both noisy and clean datasets, including 1) **FFT** which updates the entire model parameters, 2) **VPT** which fixes pre-trained model parameters and prepends a small subset of extra learnable parameters to the visual encoder during fine-tuning, and 3) **VLPT** (Vision-Language Prompt Tuning) which integrates both visual and textual learnable prompts into the fixed pre-trained model for fine-tuning. For FFT and VPT, we learn an additional linear classifier, while VLPT directly utilizes the learned textual prompts for classification. We present our three primary findings in the following.

**VPT benefits representation learning in the presence of massive noisy labels.** Intuitively, utilizing FFT for CLIP adaptation yields improved performance by leveraging its substantial capacity to incorporate task-specific representations. However, the efficacy of FFT diminishes notably when applied to datasets containing noisy labels, as shown in Figure 1a. This decline is attributed to the pronounced distortion of representations with an escalating noise ratio [1]. In contrast, VPT benefits representation learning when adapting CLIP on noisy data, particularly under high levels of noise. As only a small set of parameters is introduced, VPT effi-

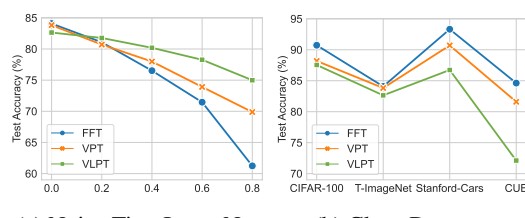

(a) Noisy Tiny-ImageNet    (b) Clean Datasets

Figure 1: Comparison of different fine-tuning methods under (a) various ratios of noisy labels and (b) clean datasets.

ciently retains the generalization ability from image-text pre-training while enhancing classification performance on downstream tasks, making it a robust and effective fine-tuning approach for handling noisy downstream datasets.

**Textual classifier is robust to noisy labels.** Classification with learnable textual prompts (e.g., CoOp [53]) leverages the multi-modal information in pre-trained vision-language models and enhances the alignment of visual and textual representations on downstream tasks. Recent literature has substantiated its robustness with a frozen visual encoder in the context of few-shot learning [41]. From Figure 1a, we observe that VLPT with a textual classifier (green line) consistently surpasses VPT with a traditional linear classifier for classification (orange line), especially under severe noise. The improvement in performance across diverse noise ratios further affirms the robustness of learnable textual prompts in mitigating the impact of label noise for model adaptation.

**FFT enhances visual recognition on clean datasets.** Although previous findings show that FFT is more susceptible to noisy labels compared to VPT and VLPT, Figure 1b demonstrates its enhanced discriminative ability when training data is clean. This superiority is particularly significant on fine-grained datasets such as Stanford-Cars and CUB-200-2011, with an average improvement of 2.81% against VPT. It is important to note that VLPT exhibits the worst performance on clean datasets. This is primarily due to the implicit regularization of pre-trained textual information when tuning the context of textual prompts, as explained in the recent study [41].

## 4 The Denoising Fine-tuning Framework

Based on the findings in Section 3.2, we now present a novel denoising fine-tuning framework, DEFT, which comprises two pivotal phases. In the first phase, DEFT learns dual textual prompts to separate clean and noisy samples while adapting the visual encoder utilizing PEFT methods. In the second phase, DEFT re-adapts the pre-trained model using FFT, leveraging the curated clean samples to further boost visual recognition performance. Figure 2 illustrates our proposed framework.

### 4.1 Identifying Noisy Labels with Dual Prompts

Given the empirically demonstrated robustness of textual prompts to label noise, we intend to harness this property to identify corrupted samples in noisy downstream datasets. The initial strategy is to select a subset of samples with pronounced similarity between their visual and textual representations and designate them as clean. However, this method relies on either prior knowledge of the noise ratio or the manual setting of thresholds, which restricts its practical applicability. Inspired by recent works [33, 14, 36], we utilize dual textual prompts instead of a single prompt to overcome these limitations. Concretely, we design a class-specific pair of *positive* and *negative* prompts for the textual encoder,

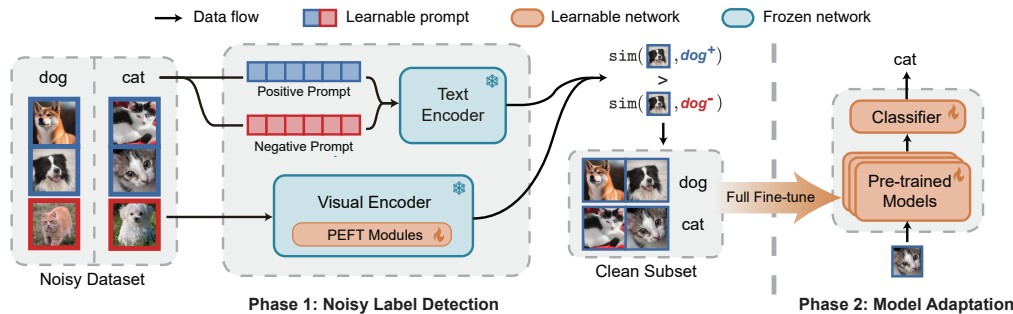

Figure 2: Illustration of the proposed DEFT framework. Left: We identify noisy labels with learnable dual textual prompts and improve image-text alignment by optimizing PEFT modules. Right: Adapt pre-trained models using FFT on selected clean samples.

denoted as $\text{prompt}_k^+$ and $\text{prompt}_k^-$, respectively:

$$\text{prompt}_k^+ = [V]_1^+[V]_2^+[V]_3^+...[V]_M^+[\text{CLS}]_k \tag{2}$$

$$\text{prompt}_k^- = [V]_1^-[V]_2^-[V]_3^-...[V]_M^-[\text{CLS}]_k \tag{3}$$

Here, $[V]_m$ is the word vector of context (with dimension 512 in CLIP), $M$ is the length of context, and $[\text{CLS}]_k$ is the $k$-th provided category name. The positive prompt aims to uncover distinguishable features by maximizing the similarity between the image features and their corresponding text features, while the negative prompt serves as a learnable sample-dependent similarity threshold to select clean data. Specifically, the *learnable threshold* $\phi_i$ for the $i$-th training sample with label $y_i = k$ is formulated as:

$$\phi_i = \text{sim}(\boldsymbol{I}_i, \boldsymbol{T}_k^-) \tag{4}$$

which represents the cosine similarity between the extracted image features and negative text features. Based on this threshold, the clean subset of dataset $\mathcal{D}$ can be constructed as follows:

$$\mathcal{D}^{\text{clean}} = \{(\boldsymbol{x}_i, y_i) \mid \text{sim}(\boldsymbol{I}_i, \boldsymbol{T}_k^+) > \phi_i \text{ and } y_i = k\} \tag{5}$$

The proposed selection strategy surpasses the conventional loss-based approaches in two aspects: 1) It is more practical by utilizing data-driven thresholds thus eliminating the requirement for prior knowledge like noise ratio, and 2) the integration of text modality enhances its robustness to label noise, making it capable of identifying challenging hard noise. Nevertheless, the primary dilemma lies in the optimization of positive and negative prompts using noisy downstream datasets.

## 4.2 Optimization for Noisy Label Detector

It can be an intricate task to optimize the threshold $\phi_i$ in an end-to-end manner since it is indirectly involved in the forward propagation. However, we can still optimize the associated parameters by formulating the clean probability of the $i$-th image with $y_i = k$ as:

$$p_{ik}^{\text{clean}} = \frac{\exp(\text{sim}(\boldsymbol{I}_i, \boldsymbol{T}_k^+)/\tau)}{\exp(\text{sim}(\boldsymbol{I}_i, \boldsymbol{T}_k^+)/\tau) + \exp(\text{sim}(\boldsymbol{I}_i, \boldsymbol{T}_k^-)/\tau)} \tag{6}$$

Obviously, selecting samples with the clean probability $p_{ik}^{\text{clean}} > 0.5$ is equivalent to the specified criteria in Eq. (5). In this way, the original threshold-based selection approach can be transformed into $K$ binary classification tasks to determine whether a sample is clean or not. Concretely, given a sample $(\boldsymbol{x}_i, y_i)$ from noisy datasets $\mathcal{D}$, we can employ the dual prompts of class $y_i$ as a binary classifier to identify label noise. If the classifier outputs a positive prediction of $\boldsymbol{x}_i$, it is categorized as a clean sample.

To optimize the noisy label detector, we need to construct positive and negative training samples for binary classification, i.e., images with correctly and wrongly assigned labels. Inspired by the prior work on negative learning [21], we designate each image with a randomly picked complementary label $\bar{y}$ to form negative samples. For positive samples, we initiate with the original supervision in

$\mathcal{D}$ and refine it with pseudo labels $\hat{y}$ generated by Eq. (1). The training loss is then formulated as follows:

$$\mathcal{L}_{dp} = \frac{1}{N} \sum_{i=1}^{N} \ell_{nll}(\boldsymbol{p}_i^{\text{clean}}, \hat{y}) + \ell_{nll}(1 - \boldsymbol{p}_i^{\text{clean}}, \bar{y}) \tag{7}$$

where $\boldsymbol{p}_i^{\text{clean}} = \{p_{ik}^{\text{clean}}\}_{k=1}^{K}$ is the clean probability and $\ell_{nll}(\cdot, \cdot)$ is the negative log-likelihood loss where $\ell_{nll}(\boldsymbol{p}_i, y) = -\log p_{iy}$.

Additionally, to enhance task-specific representations of the noisy label detector, we harness PEFT for the adaptation of the visual encoder, considering its robustness under massive noisy labels compared with FFT, as per our previous finding (see Figure 1a). This is achieved by maximizing the alignment between image embeddings and their corresponding positive textual features:

$$\mathcal{L}_{sim} = -\frac{1}{N} \sum_{i=1}^{N} \log \frac{\exp(\texttt{sim}(\boldsymbol{I}_i, \boldsymbol{T}_i^+)/\tau)}{\sum_{k=1}^{K} \exp(\texttt{sim}(\boldsymbol{I}_i, \boldsymbol{T}_k^+)/\tau)} \tag{8}$$

Note that $\mathcal{L}_{sim}$ is exclusively computed on $\mathcal{D}^{\text{clean}}$ after the warm-up stage to further eliminate the impact of label noise on representations. With the objective $\mathcal{L} = \mathcal{L}_{dp} + \mathcal{L}_{sim}$, DEFT effectively sieves out noisy samples with learned thresholds while uncovering distinguishable visual representations on downstream tasks.

### 4.3 Model Adaptation using Clean Data

Although the learned positive textual prompt can be readily employed for classification, its performance may be suboptimal on curated clean datasets, as demonstrated in our previous finding (see Figure 1b). To mitigate this problem, we introduce the model adaptation phase which learns a linear classifier using selected clean samples, i.e., $\mathcal{D}^{\text{clean}}$. Moreover, our findings also indicate that FFT outperforms PEFT on clean datasets, so we remove the PEFT modules and fully fine-tune the pre-trained model parameters for model adaptation. We minimize the classic cross-entropy loss $\ell_{ce}(\boldsymbol{x}, y) = -\log \frac{\exp(z_y)}{\sum_{k=1}^{K} \exp(z_k)}$, where $\boldsymbol{z} = \{z_k\}_{k=1}^{K}$ represents the logits predicted by the linear classifier. It is noteworthy that with the clean subset of training data constructed by Eq. (5), the second phase can be applied universally to a wide range of pre-trained models, regardless of their backbones. The versatility of our approach is demonstrated in the experiments.

## 5 Experiment

We now proceed to the experiments, demonstrating the advantages of our approach in noisy label detection and image classification on both synthetic and real-world datasets.

### 5.1 Experimental Settings

**Synthetic Datasets** We first evaluate the performance of DEFT on four image classification benchmarks by synthesizing noisy labels with varying noise types and ratios. Specifically, we conduct experimental analyses on widely-used CIFAR-100 [23] and Tiny-ImageNet [42], as well as two fine-grained datasets Stanford-Cars [22] and CUB-200-2011 [34]. Given the noise transition matrix $T \in [0,1]^{K \times K}$ with $T_{ij}$ denoting the probability of the ground-truth $y^* = i$ being flipped into a corrupted label $y = j$, we introduce two types of label noise: 1) $T_{ij} = p(y = j \mid y^* = i)$, where the corruption probability is assumed to be only dependent on the true label, e.g., the *symmetric* noise generated by replacing $y^*$ with a random label for a given proportion of training samples [10, 25], and 2) $T_{ij} = p(y = j \mid y^* = i, x)$, which is a more realistic assumption considering the impact of instance $x$ in the label corruption process, e.g., the *instance-dependent* noise proposed in [45]. We conduct extensive experiments on each dataset with noise ratio $r \in \{0.2, 0.4, 0.6\}$ for symmetric noise and $r \in \{0.2, 0.3, 0.4\}$ for instance-dependent noise.

**Real-World Datasets** We further examine the performance of DEFT on three real-world noisy label datasets: 1) CIFAR-100N [39] ($r \approx 0.4$) is a variant of CIFAR-100 with real-world human annotations from Amazon's Mechanical Turk, 2) Clothing1M [47] ($r \approx 0.4$) is a large-scale dataset consisting of 1 million clothing images of 14 classes collected from online shopping websites, and

| Method | Sym. 0.2 | | Sym. 0.4 | | Sym. 0.6 | | Ins. 0.2 | | Ins. 0.3 | | Ins. 0.4 | |
|---|---|---|---|---|---|---|---|---|---|---|---|---|
| | Prec. | Rec. | Prec. | Rec. | Prec. | Rec. | Prec. | Rec. | Prec. | Rec. | Prec. | Rec. |
| CIFAR-100 | | | | | | | | | | | | |
| Label-match | 99.83 | 63.62 | 99.61 | 63.85 | 99.31 | 63.52 | 99.93 | 63.65 | 99.85 | 63.72 | 99.81 | 63.69 |
| Small-loss | 97.24 | 96.79 | 95.68 | 94.49 | 92.93 | 90.68 | 95.20 | 95.46 | 94.00 | 92.53 | 90.33 | 89.85 |
| DEFT (ours) | **99.51** | **97.77** | **98.75** | **97.91** | **97.04** | **97.27** | **98.47** | **97.88** | **96.32** | **97.63** | **94.08** | **95.28** |
| Δ | ↑ 2.27 | ↑ 0.98 | ↑ 3.07 | ↑ 3.42 | ↑ 4.11 | ↑ 6.59 | ↑ 3.27 | ↑ 2.42 | ↑ 2.32 | ↑ 5.10 | ↑ 3.75 | ↑ 5.43 |
| Tiny-ImageNet | | | | | | | | | | | | |
| Label-match | 99.92 | 60.81 | 99.83 | 60.79 | 99.50 | 60.66 | 99.91 | 60.58 | 99.84 | 60.53 | 99.76 | 60.47 |
| Small-loss | 97.25 | **96.93** | 95.33 | 94.48 | 92.63 | 90.89 | 94.74 | 95.17 | 93.66 | 92.35 | 90.41 | 89.71 |
| DEFT (ours) | **99.50** | 96.00 | **98.78** | **95.97** | **97.21** | **95.44** | **99.21** | **96.21** | **97.80** | **95.80** | **95.45** | **95.77** |
| Δ | ↑ 2.25 | ↓ 0.93 | ↑ 3.45 | ↑ 1.49 | ↑ 4.58 | ↑ 4.55 | ↑ 4.47 | ↑ 1.04 | ↑ 4.14 | ↑ 3.45 | ↑ 5.04 | ↑ 6.06 |
| Stanford-Cars | | | | | | | | | | | | |
| Label-match | 99.97 | 60.34 | 99.86 | 60.27 | 99.70 | 60.71 | 99.85 | 60.34 | 99.82 | 60.32 | 99.80 | 60.25 |
| Small-loss | 96.92 | 96.56 | 93.71 | 93.21 | 89.46 | 87.79 | 96.94 | 97.78 | 96.72 | 95.96 | 95.25 | 94.48 |
| DEFT (ours) | **98.72** | **99.56** | **98.98** | **98.56** | **98.58** | **95.62** | **99.02** | **99.09** | **98.96** | **98.15** | **98.75** | **97.71** |
| Δ | ↑ 1.80 | ↑ 3.00 | ↑ 5.27 | ↑ 5.35 | ↑ 9.12 | ↑ 7.83 | ↑ 2.08 | ↑ 1.31 | ↑ 2.24 | ↑ 2.19 | ↑ 3.50 | ↑ 3.23 |
| CUB-200-2011 | | | | | | | | | | | | |
| Label-match | 99.92 | 53.26 | 99.74 | 53.13 | 99.46 | 53.02 | 99.96 | 53.39 | 99.96 | 53.32 | 99.74 | 53.69 |
| Small-loss | 96.74 | 96.32 | 93.69 | 92.84 | 84.10 | 82.01 | 96.91 | 97.33 | 96.49 | 95.59 | 93.98 | 93.96 |
| DEFT (ours) | **99.04** | **97.01** | **96.76** | **95.60** | **93.88** | **96.43** | **99.15** | **97.45** | **97.93** | **96.85** | **96.03** | **97.11** |
| Δ | ↑ 2.30 | ↑ 0.69 | ↑ 3.07 | ↑ 2.76 | ↑ 9.78 | ↑ 14.42 | ↑ 2.24 | ↑ 0.12 | ↑ 1.44 | ↑ 1.26 | ↑ 2.05 | ↑ 3.15 |

Table 1: On each dataset, we compare the Precision (%) and Recall (%) of DEFT with CLIP label-match and small-loss to evaluate the clean sample selection performance. Δ is the difference between the performance of DEFT and small-loss.

3) WebVision [26] ($r \approx 0.2$) contains 2.4 million images crawled from Flickr and Google using the 1,000 concepts in ImageNet ILSVRC12. Following previous works [25, 19, 24], we experiment on the first 50 classes of the Google image subset.

**Evaluation Metrics**   We use precision and recall metrics to evaluate the selection of clean samples. A higher precision indicates a greater proportion of actual clean samples in $\mathcal{D}^{\text{clean}}$, while a higher recall means that more clean samples are identified from the noisy dataset. For image classification, We report the "best" test accuracy across all training iterations and the "last" test accuracy at the end of training.

**Implementation Details**   We adopt the pre-trained CLIP [29] with the Transformer as the text encoder and the ViT-B/16 as the image encoder. We use the SGD optimizer with a momentum of 0.9, a weight decay of $5\times10^{-4}$, and a batch size of 64. We run 10 epochs for both the noisy label detection phase and the model adaptation phase with learning rates $3\times10^{-2}$ and $5\times10^{-4}$, respectively. In the noisy label detection phase, we employ VPT [18] and CoOp [53] to adapt visual encoder and textual encoder respectively, and perform model warm-up for 1 epoch on all datasets. When identifying noisy labels on real-world datasets, we augment the condition in Eq. (5) with a constraint that the training label should be consistent with its predicted label. This is due to the existence of more complex noise patterns in real-world tasks. All experiments are conducted on a single NVIDIA GeForce RTX 3090.

## 5.2   Performance for Noisy Label Detection

We conduct a thorough evaluation of DEFT to justify its effectiveness in detecting noisy labels. For this purpose, we simulate various types and ratios of label noise on four benchmark datasets. We compare our approach with two other sample selection strategies: 1) **Label-match** strategy, where a training sample is deemed clean if its given label matches that assigned by the CLIP zero-shot classifier, and 2) **Small-loss** strategy, which selects a proportion (noise ratio) of confident samples in each mini-batch, as commonly used in previous studies [10, 25, 19]. The precision and recall of sample selection are reported in Table 1. The results show that our proposed noisy label detector outperforms the compared strategies, with significant improvements observed in all dataset settings. The trivial label-match strategy tends to be conservative, leading to low selection recall of clean samples. In contrast, the small-loss strategy and DEFT achieve a better trade-off between precision and recall, making better use of training samples. Moreover, leveraging the informative multi-model information in the pre-trained vision-language model, DEFT surpasses the small-loss strategy

|  | Method | *Sym.* 0.2 | *Sym.* 0.4 | *Sym.* 0.6 | *Ins.* 0.2 | *Ins.* 0.3 | *Ins.* 0.4 |
|---|---|---|---|---|---|---|---|
| | | | | CIFAR-100 | | | |
| FFT | CE | 86.71 / 86.70 | 84.06 / 82.60 | 81.05 / 77.45 | 87.30 / 87.18 | 84.60 / 83.64 | 78.41 / 75.66 |
| | ELR | 86.53 / 86.53 | 83.66 / 83.66 | 78.34 / 78.34 | 86.61 / 86.61 | 85.89 / 85.89 | **85.78 / 85.78** |
| | SCE | 86.82 / 86.82 | 83.84 / 83.84 | 78.90 / 77.71 | 86.61 / 86.61 | 83.99 / 83.20 | 80.06 / 73.45 |
| | GMM | 88.49 / 88.49 | 87.21 / 87.21 | 85.22 / 85.20 | 88.44 / 88.44 | 87.95 / 87.95 | 82.14 / 82.11 |
| DEFT | Ours | **89.38 / 89.35** | **88.17 / 88.11** | **85.81 / 85.72** | **89.38 / 89.35** | **88.68 / 88.68** | 85.75 / 85.74 |
| | | | | Tiny-ImageNet | | | |
| FFT | CE | 81.77 / 81.08 | 76.53 / 76.52 | 73.17 / 71.46 | 80.75 / 80.71 | 78.83 / 78.57 | 74.80 / 74.08 |
| | ELR | 79.40 / 79.40 | 77.13 / 77.13 | 73.74 / 73.74 | 79.98 / 79.98 | 77.13 / 77.13 | 73.74 / 73.74 |
| | SCE | 79.23 / 79.23 | 76.24 / 76.18 | 71.76 / 70.62 | 78.96 / 78.90 | 77.80 / 77.54 | 74.47 / 73.25 |
| | GMM | 81.91 / 81.88 | 80.37 / 80.37 | 43.47 / 43.47 | 81.84 / 81.79 | 81.26 / 81.26 | 79.01 / 79.01 |
| DEFT | Ours | **82.91 / 82.91** | **82.48 / 82.37** | **80.60 / 80.59** | **83.37 / 83.33** | **82.69 / 82.65** | **80.52 / 80.49** |
| | | | | Stanford-Cars | | | |
| FFT | CE | 89.75 / 89.74 | 85.10 / 84.89 | 71.70 / 71.55 | 89.13 / 89.06 | 85.94 / 85.92 | 80.59 / 80.59 |
| | ELR | 86.61 / 86.61 | 76.98 / 76.98 | 61.58 / 61.58 | 84.40 / 84.40 | 83.11 / 83.11 | 75.97 / 75.84 |
| | SCE | 91.11 / 91.11 | 87.73 / 87.45 | 79.09 / 79.09 | 90.34 / 90.34 | 87.35 / 86.23 | 83.50 / 80.69 |
| | GMM | 90.10 / 90.08 | 83.14 / 83.10 | 56.90 / 56.90 | 88.15 / 88.10 | 85.39 / 85.33 | 78.76 / 78.72 |
| DEFT | Ours | **92.13 / 92.12** | **90.75 / 90.75** | **85.72 / 85.45** | **92.19 / 92.15** | **90.77 / 90.77** | **89.74 / 89.68** |
| | | | | CUB-200-2011 | | | |
| FFT | CE | 80.76 / 80.76 | 73.09 / 72.87 | 55.42 / 55.21 | 80.36 / 80.25 | 75.80 / 75.53 | 69.62 / 69.62 |
| | ELR | 77.70 / 77.70 | 68.26 / 68.26 | 50.17 / 49.88 | 78.32 / 78.32 | 73.16 / 73.08 | 63.57 / 63.34 |
| | SCE | 82.81 / 82.74 | 78.12 / 77.87 | 63.31 / 63.31 | 81.91 / 81.91 | 78.31 / 78.03 | 71.25 / 70.95 |
| | GMM | 75.79 / 75.73 | 64.39 / 64.38 | 42.84 / 42.84 | 75.73 / 75.65 | 69.95 / 69.95 | 56.13 / 55.80 |
| DEFT | Ours | **83.05 / 83.03** | **79.24 / 79.13** | **73.08 / 73.08** | **82.53 / 82.50** | **81.39 / 81.39** | **79.34 / 79.24** |

Table 2: Test accuracy (%) on synthetic datasets with *symmetric* and *instance-dependent* label noise.

| Dataset | CE | ELR | SCE | GMM | RoLT | UNICON | LongReMix | ProMix | DEFT (Ours) |
|---|---|---|---|---|---|---|---|---|---|
| CIFAR-100N | 72.41 | 72.83 | 72.52 | 76.06 | 75.91 | 77.68 | 73.94 | 75.97 | **79.04** |
| Clothing1M | 69.75 | 72.14 | 70.49 | 70.03 | 70.46 | 70.38 | 70.62 | 70.71 | **72.44** |
| WebVision | 84.64 | 79.32 | 82.88 | 84.88 | 84.12 | 84.56 | 84.96 | 84.44 | **85.12** |

Table 3: Test accuracy (%) on datasets with real-world label noise.

in detecting noisy labels, particularly under severe noise settings and fine-grained classification tasks. For instance, in Tiny-ImageNet with 60% symmetric noise, DEFT demonstrates significant enhancements of 4.58% and 4.55% in precision and recall, respectively, as well as an improvement of 9.78% and 14.42% in CUB-200-2011. Additionally, DEFT reduces the need for prior knowledge of noise ratios, making it a practical and effective approach to detect label noise in real-world tasks.

## 5.3 Performance for Image Classification

We evaluate the performance of DEFT in image classification tasks against four baselines, including CE (Cross-Entropy loss), SCE [37], ELR [27], and GMM [25]. To further verify the effectiveness of our method on real-world datasets, we also make comparisons with some recent sample selection-based works including RoLT[40], UNICON[19], LongReMix[4], and ProMix[46]. We utilize FFT for adapting the pre-trained CLIP model for all approaches. More details can be found in the Appendix.

**Results on Synthetic Datasets**    Table 2 presents the results on four synthetic noisy datasets. Compared with CE, the adoption of noise-robust loss functions such as ELR and SCE improves the classification performance on CIFAR-100 and Tiny-ImageNet under certain noisy label settings, e.g., ELR achieves the best performance on CIFAR-100 with 40% instance-dependent noise. However, these methods are not always effective and may even worsen the performance under varying types and ratios of noise, compared to the sample selection paradigm like GMM and DEFT. While GMM outperforms ELR and SCE in most cases, its performance degrades dreadfully on fine-grained datasets. Nevertheless, DEFT retains the most robust performance and generally advances all compared methods, especially on Stanford-Cars and CUB-200-2011. For example, DEFT boosts the test accuracy

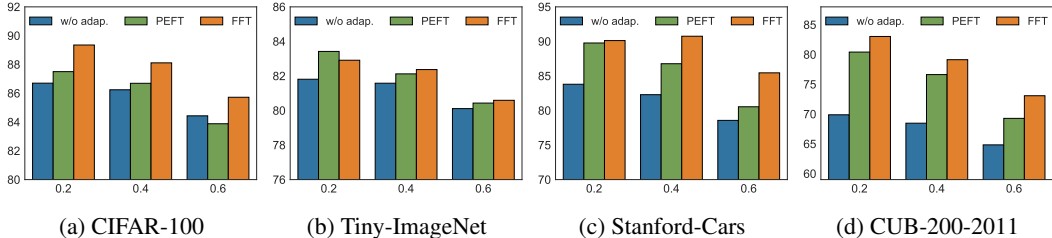

| (a) CIFAR-100 | (b) Tiny-ImageNet | (c) Stanford-Cars | (d) CUB-200-2011 |

Figure 3: Ablation studies. We report the test accuracy across varying noise ratios for the following variants: 1) **w/o adap.**: DEFT without the model adaptation phase, 2) **PEFT**: use PEFT for model adaptation phase, and 3) **FFT**: use FFT for model adaptation phase.

| Architecture | CE | GCE | ELR | TURN | DEFT (Ours) |
|---|---|---|---|---|---|
| ResNet-50 [12] | 66.02 | 66.19 | 66.19 | 66.31 | **70.82** |
| MAE-ViT-B [11] | 61.31 | 60.80 | 61.51 | 61.96 | **65.23** |
| ViT-B/16 [5] | 68.98 | 69.74 | 68.73 | **70.28** | 69.84 |
| ConvNeXt-T [28] | 68.80 | 68.92 | 68.52 | 69.53 | **71.68** |

Table 4: Test accuracy (%) using various pre-trained models on Clothing1M. Partial results are sourced from [1]. The best results across all methods are highlighted in bold, with the second-best results indicated by underscores.

by an average of 4.34% on Stanford-Cars compared to the best results of comparison methods. The results demonstrate the effectiveness of our proposed approach in tackling both symmetric and instance-dependent label noise.

**Results on Real-world Datasets**   The results reported in Table 3 demonstrate the superiority of our method on real-world datasets, as DEFT surpasses other methods by a significant margin on all datasets. Notably, while most compared methods show little improvement on Clothing1M due to its fine-grained nature, both ELR and DEFT demonstrate substantial improvement. However, ELR exhibits a performance decrease on WebVision. Moreover, directly fine-tuning pre-trained CLIP models with cross-entropy (CE) achieves competitive results on WebVision, yet DEFT elevates this performance further, showcasing its ability to combat label noise in practical situations.

## 5.4   Further Analyses

**Necessity of Model Adaptation**   In this experiment, we emphasize the necessity of the model adaptation phase in DEFT. Figure 3 demonstrates the test accuracy of DEFT without the model adaptation phase (w/o adap.), adaptation utilizing PEFT and FFT. Results show that fully fine-tuning the model on selected clean samples yields the best classification performance, especially on fine-grained datasets, such as Stanford-Cars and CUB-200-2011. The results unveil two primary insights: 1) the noisy label detector in DEFT exhibits strong capabilities in detecting label noise, yielding a nearly clean subset from noisy downstream datasets, and 2) employing FFT for model adaptation is more effective in mitigating significant domain shifts between downstream data and pre-training data, particularly evident in fine-grained datasets.

**DEFT for Various Pre-trained Models**   It is noteworthy that DEFT can seamlessly integrate with various pre-trained visual backbones during the model adaptation phase. To demonstrate the versatility of our proposed denoising fine-tuning framework, we conduct experiments on Clothing1M and apply FFT to a range of pre-trained models, including ViT-B/16 [5], ResNet-50 [12], ConvNeXt-T [28], and MAE-ViT-B [11]. The results are presented in Table 4, where GCE [51] employs a noise-robust loss function akin to SCE. As a recent work, TURN [1] is an improved version of GMM, which applies linear probing to initialize the classifier and then uses FFT for adaptation. Compared with previous methods, DEFT exhibits the best performance on average. In particular, DEFT outperforms TURN by 4.51% and 3.27% when adopting ResNet-50 and MAE-ViT-B as the target pre-trained models, respectively.

# 6  Conclusion

In this work, we delve into a new landscape for learning with noisy labels, departing from the classic single-modal toward a multi-modal regime. By learning dual textual prompts, we construct a new noisy label detection approach, which offers several compelling advantages: robust to various types of label noise, generalizable to many pre-trained models, and does not require the dynamics of training samples. We investigate the effectiveness of DEFT on a wide range of synthetic and real-world datasets, showing its superior performance in both noisy label detection and image classification tasks. Lastly, we demonstrate the advantage of parameter-efficient fine-tuning over full fine-tuning over noisy label detection. We hope our work will inspire future research toward multi-modal noisy label detection.

## Acknowledgments and Disclosure of Funding

This work was supported by the National Science Foundation of China (62206049, 62225602), the Key Program of Jiangsu Science Foundation (BK20243012), and the Big Data Computing Center of Southeast University. We would like to thank anonymous reviewers for their constructive suggestions.

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

# A  Appendix

## A.1   Additional Results on Noisy Label Detection

**Impact of CLIP Backbones**   Table 1 has demonstrated the superiority of DEFT in detecting noisy labels based on CLIP ViT-B/16. To evaluate the impact of different visual backbones on noise detection, we re-implement all methods using CLIP ViT-B/32 as the backbone and report the results in Table 5. Notably, while the small-loss strategy demonstrates comparable performance on noisy CIFAR-100 and Tiny-ImageNet, its efficacy significantly diminishes on fine-grained datasets such as Stanford-Cars and CUB-200-2011, which exhibits its sensitivity to visual backbones. In contrast, DEFT maintains consistently superior performance on both ViT-B/16 and ViT-B/32, underscoring its resilience across different pre-trained visual backbones.

**Results under Severe Noise**   We conduct experiments on synthetic noisy datasets with significant label noise to further validate the robustness of the proposed noisy label detector. It is noteworthy that two adjustments are implemented in order to train a more effective model: 1) the learning rate is adjusted from $3 \times 10^{-2}$ to $1 \times 10^{-2}$, and 2) For DEFT, a small weight (specifically, 0.25) is introduced to the positive loss component in Eq. (7) to mitigate the impact of noisy pseudo labels in high levels of noise. Table 6 reports the precision, recall, and their harmonic mean (F1-score) for noise detection, where DEFT exhibits a robust capability in identifying severe noise. For example, a remarkable improvement in the F1-score is achieved on CUB-200-2011. These results strongly reaffirm the superior performance of DEFT across various noise ratios.

**GMM for Noisy Label Detection**   DivideMix [25] introduces the use of a Gaussian Mixture Model (GMM) for the dynamic separation of clean and noisy samples with an adjustable threshold. Table 7 summarizes the performance of GMM and DEFT in detecting label noise across diverse datasets and noise settings. To fine-tune the pre-trained model for GMM, we independently apply PEFT and FFT. The results reveal that DEFT consistently exhibits superior noise detection capability, particularly evident in fine-grained datasets with high noise ratios, such as Stanford-Cars with 60% symmetric noise and CUB-200-2011 with 40% instance-dependent noise. Intriguingly, implementing GMM with PEFT yields better noise detection when compared to utilizing FFT. This difference can be attributed to the potentially unstable training process of FFT, where the loss value may undergo significant fluctuations due to an abundance of tunable parameters.

**Noisy Label Detection on CIFAR-100N**   The CIFAR-100N dataset [39], which serves as a real-world benchmark by providing CIFAR-100 with human-annotated noisy labels, allows us to evaluate

| Method | Sym. 0.2 | | Sym. 0.4 | | Sym. 0.6 | | Sym. 0.2 | | Sym. 0.3 | | Sym. 0.4 | |
|---|---|---|---|---|---|---|---|---|---|---|---|---|
| | Prec. | Rec. | Prec. | Rec. | Prec. | Rec. | Prec. | Rec. | Prec. | Rec. | Prec. | Rec. |
| CIFAR-100 | | | | | | | | | | | | |
| Label-match | 99.82 | 57.91 | 99.51 | 58.13 | 99.10 | 58.03 | 99.87 | 57.90 | 99.82 | 57.86 | 99.72 | 57.84 |
| Small-loss | 97.32 | 96.86 | 95.43 | 94.23 | 92.78 | 90.46 | 94.78 | 95.00 | 93.33 | 91.83 | 89.61 | 89.12 |
| DEFT (ours) | **99.41** | **97.88** | **98.51** | **97.80** | **96.10** | **97.74** | **98.89** | **97.89** | **97.00** | **97.82** | **93.27** | **97.76** |
| Δ | ↑ 2.09 | ↑ 1.02 | ↑ 3.08 | ↑ 3.57 | ↑ 3.32 | ↑ 7.28 | ↑ 4.11 | ↑ 2.89 | ↑ 3.67 | ↑ 5.99 | ↑ 3.66 | ↑ 8.64 |
| Tiny-ImageNet | | | | | | | | | | | | |
| Label-match | 99.88 | 58.02 | 99.80 | 58.11 | 99.47 | 58.26 | 99.89 | 58.00 | 99.82 | 58.04 | 99.74 | 57.93 |
| Small-loss | 97.20 | **96.88** | 95.44 | 94.56 | 92.73 | 90.89 | 94.72 | 95.14 | 93.36 | 92.05 | 89.83 | 89.16 |
| DEFT (ours) | **99.45** | 95.31 | **98.52** | **95.79** | **96.08** | **95.88** | **99.04** | **95.71** | **98.08** | **95.88** | **96.16** | **95.76** |
| Δ | ↑ 2.25 | ↓ 1.57 | ↑ 3.08 | ↑ 1.23 | ↑ 3.35 | ↑ 4.99 | ↑ 4.32 | ↑ 0.57 | ↑ 4.72 | ↑ 3.83 | ↑ 6.33 | ↑ 6.60 |
| Stanford-Cars | | | | | | | | | | | | |
| Label-match | 99.88 | 50.14 | 99.51 | 50.14 | 99.65 | 51.28 | 99.81 | 49.84 | 99.72 | 49.79 | 99.55 | 49.90 |
| Small-loss | 96.98 | 96.60 | 86.51 | 85.43 | 41.23 | 39.92 | 95.02 | 95.75 | 78.25 | 77.31 | 63.71 | 62.67 |
| DEFT (ours) | **99.17** | **99.23** | **97.40** | **99.31** | **94.66** | **97.68** | **98.68** | **98.95** | **97.75** | **98.96** | **95.62** | **99.33** |
| Δ | ↑ 2.19 | ↑ 2.63 | ↑ 10.89 | ↑ 13.88 | ↑ 53.43 | ↑ 57.76 | ↑ 3.66 | ↑ 3.20 | ↑ 19.50 | ↑ 21.65 | ↑ 31.91 | ↑ 36.66 |
| CUB-200-2011 | | | | | | | | | | | | |
| Label-match | 99.83 | 49.24 | 99.67 | 49.72 | 99.75 | 49.38 | 99.79 | 50.07 | 99.71 | 50.13 | 99.61 | 50.36 |
| Small-loss | 96.12 | 95.68 | 87.41 | 86.35 | 58.65 | 56.99 | 94.42 | 94.85 | 86.66 | 85.74 | 77.08 | 76.77 |
| DEFT (ours) | **98.62** | **97.56** | **95.15** | **98.18** | **94.32** | **95.85** | **98.18** | **97.00** | **93.57** | **93.05** | **95.35** | **97.41** |
| Δ | ↑ 2.50 | ↑ 1.88 | ↑ 7.74 | ↑ 11.83 | ↑ 35.67 | ↑ 38.86 | ↑ 3.76 | ↑ 2.15 | ↑ 6.91 | ↑ 7.31 | ↑ 18.27 | ↑ 20.64 |

Table 5: Precision and recall of noisy label detection with CLIP ViT-B/32 as the visual backbone.

| Method | Sym. 0.8 | | | Ins. 0.5 | | |
|---|---|---|---|---|---|---|
| | Prec. | Rec. | F1. | Prec. | Rec. | F1. |
| CIFAR-100 | | | | | | |
| Label-match | 97.94 | 63.68 | 77.18 | 99.69 | 63.86 | 77.85 |
| Small-loss | 90.31 | 83.69 | 86.87 | 85.37 | 86.45 | 85.91 |
| DEFT (ours) | **95.25** | **91.60** | **93.39** | **92.83** | **93.08** | **92.95** |
| Tiny-ImageNet | | | | | | |
| Label-match | 98.74 | 61.04 | 75.44 | 99.68 | 60.61 | 75.38 |
| Small-loss | 89.12 | 84.52 | 86.76 | 84.92 | 85.04 | 84.98 |
| DEFT (ours) | **95.97** | **89.68** | **92.72** | **96.19** | **90.86** | **93.45** |
| Stanford-Cars | | | | | | |
| Label-match | 98.62 | 60.52 | 75.01 | 99.63 | 59.49 | 74.50 |
| Small-loss | 81.97 | 77.39 | 79.61 | 89.18 | 88.77 | 88.97 |
| DEFT (ours) | **92.78** | **95.31** | **94.03** | **95.80** | **99.02** | **97.38** |
| CUB-200-2011 | | | | | | |
| Label-match | 98.20 | 53.22 | 69.03 | 99.44 | 53.50 | 69.57 |
| Small-loss | 71.23 | 66.41 | 68.74 | 83.81 | 84.65 | 84.23 |
| DEFT (ours) | **90.12** | **89.94** | **90.03** | **95.02** | **96.47** | **95.74** |

Table 6: Precision, recall, and F1-score of noisy label detection under 80% *symmetric* noise and 50% *instance-dependent* noise.

| Method | Symmetric | | | Instance-dependent | | |
|---|---|---|---|---|---|---|
| | 0.2 | 0.4 | 0.6 | 0.2 | 0.3 | 0.4 |
| CIFAR-100 | | | | | | |
| GMM w/ FFT | 97.44 | 97.56 | **97.19** | 97.08 | **96.99** | 87.98 |
| GMM w/ PEFT | 97.82 | 97.74 | 96.94 | 97.08 | 95.86 | 91.37 |
| DEFT (ours) | **98.63** | **98.33** | 97.15 | **98.17** | 96.97 | **94.68** |
| Tiny-ImageNet | | | | | | |
| GMM w/ FFT | 94.80 | 95.03 | 65.27 | 94.36 | 94.42 | 91.66 |
| GMM w/ PEFT | 96.56 | 96.64 | 96.24 | 96.01 | 95.02 | 92.04 |
| DEFT (ours) | **97.72** | **97.35** | **96.32** | **97.69** | **96.79** | **95.61** |
| Stanford-Cars | | | | | | |
| GMM w/ FFT | 97.81 | 95.02 | 77.00 | 96.60 | 95.27 | 91.05 |
| GMM w/ PEFT | 99.10 | 98.31 | 93.22 | 98.34 | 96.58 | 92.46 |
| DEFT (ours) | **99.14** | **98.77** | **97.08** | **99.05** | **98.55** | **98.23** |
| CUB-200-2011 | | | | | | |
| GMM w/ FFT | 91.92 | 86.56 | 67.52 | 91.86 | 88.58 | 76.34 |
| GMM w/ PEFT | 97.06 | 96.57 | 91.39 | 97.27 | 94.45 | 89.20 |
| DEFT (ours) | **98.01** | **96.18** | **95.14** | **98.29** | **97.39** | **96.57** |

Table 7: F1-score (%) of noisy label detection. For GMM, we utilize both PEFT and FFT to adapt the pre-trained model.

| | Label-match | Small-loss | GMM | RoLT | UNICON | LongReMix | ProMix | DEFT (Ours) |
|---|---|---|---|---|---|---|---|---|
| Precision | 91.48 | 89.37 | 80.03 | 75.23 | 87.05 | 69.37 | 75.92 | 88.43 |
| Recall | 73.88 | 89.14 | 97.05 | 95.82 | 85.44 | 98.67 | 94.90 | 92.84 |
| F1-score | 81.74 | 89.25 | 87.72 | 84.29 | 86.24 | 81.47 | 84.36 | 90.58 |

Table 8: Noise detection results on CIFAR-100N.

| Hyperpara. | visual prompt length | | | | textual prompt length | | | | training epoch | | | |
|---|---|---|---|---|---|---|---|---|---|---|---|---|
| | 10 | 15 | 20 | 30 | 4 | 8 | 16 | 32 | 5 | 10 | 15 | 20 |
| F1-score | 98.08 | 98.17 | 98.33 | 98.30 | 93.29 | 97.14 | 97.15 | 97.31 | 96.65 | 97.15 | 97.34 | 97.38 |

Table 9: Sensitivity analysis of promt length and training epoch.

the efficacy of DEFT for noise detection in practical scenarios. The results presented in Table 8 reveal the distinctive performance characteristics of various sample selection strategies. For example, the label-match strategy demonstrates a propensity for being greedy and yields the highest precision, whereas the GMM and recent sample selection-based works tends to be conservative and thus achieve a higher recall. Nevertheless, DEFT stands out with the highest F1-score across all strategies, striking a judicious balance between precision and recall in the identification of real-world noisy labels.

**Sensitivity of Hyperparameters** We explore the sensitivity of our noise label detector to various hyperparameter settings, specifically the length of the visual/textual prompt and the number of training epochs. Table 9 presents the F1-score on CIFAR-100 with 40% symmetric noise. The results indicate that the length of the visual prompt has a marginal impact on the noisy label detection. For the textual prompt length, once the context length exceeds 8, the performance is enhanced slightly as the length increases. Additionally, variations in the number of training epochs have minimal impact on performance. In order to strike a balance between effectiveness and efficiency, we set the visual prompt length to 20, the textual prompt length to 16, and the training epoch to 10 in all experiments.

## A.2 Additional Results on Image Classification

**Robustness of Various PEFT techniques** To further substantiate the advantages of PEFT over FFT for adapting visual backbones in the presence of massive noisy labels, we conduct a comprehensive evaluation of various PEFT techniques for image classification, including 1) *Visual Prompt Tuning* (VPT) [18] which prepends learnable prompts to the input at each layer, 2) *Low-Rank Adapter* (LoRA) [13] which injects trainable rank decomposition matrices into each layer, 3) *AdaptFormer* [3] which replaces the original Multi-Layer Perceptron (MLP) block with AdaptMLP that incorporates a trainable bottleneck module, and 4) *Bias-terms Fine-tuning* (BitFit) [50] which is a sparse fine-

| | Method | *Sym.* 0.2 | *Sym.* 0.4 | *Sym.* 0.6 | *Ins.* 0.2 | *Ins.* 0.3 | *Ins.* 0.4 |
|---|---|---|---|---|---|---|---|
| | | CIFAR-100 | | | | | |
| PEFT | CE | 84.86 / 84.86 | 82.70 / 82.42 | 79.32 / 75.49 | 81.35 / 81.29 | 75.64 / 75.57 | 68.13 / 68.13 |
| | ELR | 87.17 / 87.16 | 85.95 / 85.89 | 84.29 / 84.29 | 87.05 / 87.01 | 86.93 / 86.91 | **86.58 / 86.57** |
| | SCE | 86.94 / 86.90 | 85.24 / 85.17 | 82.40 / 81.66 | 83.98 / 83.81 | 81.16 / 79.40 | 77.43 / 72.30 |
| | GMM | 86.99 / 86.95 | 85.53 / 85.51 | 84.16 / 84.16 | 86.64 / 86.64 | 85.06 / 85.06 | 80.02 / 80.02 |
| DEFT | w/ PEFT | 87.50 / 87.50 | 86.69 / 86.69 | 84.00 / 83.88 | 87.58 / 87.58 | 85.68 / 85.64 | 83.73 / 83.71 |
| | w/ FFT | **89.38 / 89.35** | **88.17 / 88.11** | **85.81 / 85.72** | **89.38 / 89.35** | **88.68 / 88.68** | 85.75 / 85.74 |
| | | Tiny-ImageNet | | | | | |
| PEFT | CE | 80.73 / 80.71 | 77.99 / 77.99 | 74.44 / 73.91 | 79.58 / 79.58 | 75.93 / 75.89 | 71.08 / 71.08 |
| | ELR | 83.01 / 83.01 | 81.81 / 81.81 | 79.34 / 79.34 | 82.93 / 82.93 | 82.28 / 82.28 | **81.74 / 81.71** |
| | SCE | 82.84 / 82.84 | 81.25 / 81.27 | 78.09 / 78.09 | 80.96 / 80.96 | 79.02 / 78.96 | 75.73 / 74.57 |
| | GMM | 82.89 / 82.87 | 81.74 / 81.72 | 80.35 / 80.35 | 81.53 / 81.53 | 81.79 / 81.79 | 78.61 / 78.61 |
| DEFT | w/ PEFT | **83.42 / 83.42** | 82.12 / 82.12 | 80.43 / 80.43 | 82.76 / 82.74 | 82.63 / 82.57 | 81.31 / 81.24 |
| | w/ FFT | 82.91 / 82.91 | **82.48 / 82.37** | **80.60 / 80.59** | **83.37 / 83.33** | **82.69 / 82.65** | 80.52 / 80.49 |
| | | Stanford-Cars | | | | | |
| PEFT | CE | 75.79 / 72.30 | 66.29 / 54.62 | 49.09 / 33.86 | 74.23 / 72.72 | 66.50 / 64.13 | 58.85 / 54.31 |
| | ELR | 87.18 / 87.18 | 81.04 / 81.00 | 70.44 / 70.44 | 86.53 / 86.53 | 84.42 / 84.33 | 80.09 / 79.93 |
| | SCE | 77.95 / 71.84 | 70.14 / 55.11 | 53.84 / 35.74 | 77.74 / 70.20 | 71.02 / 61.03 | 65.38 / 51.18 |
| | GMM | 88.97 / 88.97 | 85.91 / 85.60 | 75.82 / 75.82 | 88.05 / 88.04 | 85.51 / 85.39 | 78.61 / 78.61 |
| DEFT | w/ PEFT | 89.80 / 89.77 | 86.82 / 86.77 | 81.43 / 80.54 | 89.72 / 89.65 | 88.01 / 87.80 | 86.58 / 86.36 |
| | w/ FFT | **92.13 / 92.12** | **90.75 / 90.75** | **85.72 / 85.45** | **92.19 / 92.15** | **90.77 / 90.77** | **89.74 / 89.68** |
| | | CUB-200-2011 | | | | | |
| PEFT | CE | 69.35 / 65.22 | 54.42 / 46.91 | 43.98 / 30.77 | 66.43 / 66.12 | 59.65 / 56.35 | 52.33 / 47.26 |
| | ELR | 79.20 / 79.01 | 72.04 / 71.99 | 60.34 / 60.24 | 78.32 / 78.29 | 75.73 / 75.73 | 70.78 / 70.25 |
| | SCE | 70.12 / 65.55 | 61.84 / 45.31 | 49.55 / 29.25 | 69.97 / 62.82 | 63.55 / 54.28 | 56.68 / 45.01 |
| | GMM | 78.75 / 78.51 | 77.72 / 76.91 | 64.83 / 64.69 | 80.10 / 80.10 | 76.06 / 76.06 | 67.97 / 67.97 |
| DEFT | w/ PEFT | 80.55 / 80.41 | 76.63 / 76.63 | 69.71 / 69.28 | 80.57 / 80.50 | 77.25 / 77.11 | 74.42 / 74.20 |
| | w/ FFT | **83.05 / 83.03** | **79.24 / 79.13** | **73.08 / 73.08** | **82.53 / 82.50** | **81.39 / 81.39** | **79.34 / 79.24** |

Table 10: Test accuracy (%) on synthetic datasets with *symmetric* and *instance-dependent* label noise. For all approaches, we utilize PEFT to adapt the pre-trained CLIP model. The results of DEFT when applying FFT for adaptation are also reported.

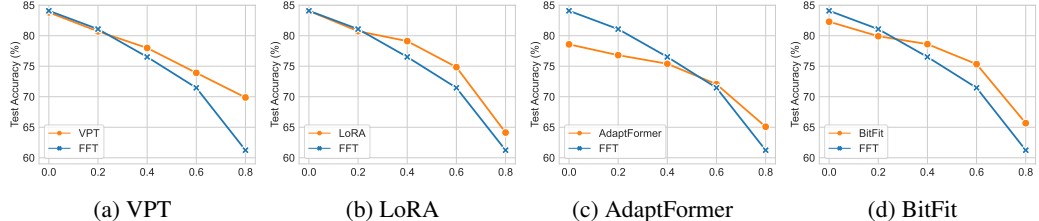

| (a) VPT | (b) LoRA | (c) AdaptFormer | (d) BitFit |
|---|---|---|---|

Figure 4: Comparison of different parameter-efficient fine-tuning techniques on Tiny-ImageNet with various ratios of noisy labels

tuning method where only the bias-terms of the model are being modified. The results are shown in Figure 4. Notably, nearly all PEFT methods outperform FFT when the noise ratio exceeds 0.4, and the performance gap becomes more pronounced with the increase in the noise ratio. These results comprehensively underscore the efficacy of PEFT over FFT in mitigating the distortion of representations when adapting visual backbones under severe noise.

**Applying PEFT for Model Adaptation** Table 10 presents the test accuracy obtained by applying PEFT to adapt the pre-trained CLIP model. There are three noteworthy observations in comparison to the results in Table 2. Firstly, previous label-noise learning methods consistently demonstrate their efficacy in handling noisy datasets compared to the vanilla methods (CE), showcasing a clear improvement in performance across various datasets and noise settings. Secondly, despite the bad performance of GMM on fine-grained datasets in Table 2, it achieves the best performance among

**Algorithm 1:** The Proposed DEFT Framework

---

1  **Input**: training dataset $\mathcal{D} = \{(\boldsymbol{x}_i, y_i)_{i=1}^N\}$, PEFT parameters $\omega$, pre-trained parameters $\theta$, warm-up epoch $T_0$, PEFT epoch $T_1$ and FFT epoch $T_2$.
   // Phase1: Learning Noisy Label Detector with PEFT
2  **for** $t = 1, 2, ..., T_0$ **do**
3     |  Warm-up the pre-trained model on noisy dataset $\mathcal{D} = \{(\boldsymbol{x}_i, y_i)_{i=1}^N\}$
4  **end**
5  **for** $t = T_0 + 1, ..., T_1$ **do**
6     |  Construct the clean subset $\mathcal{D}^{\text{clean}}$ by Eq. (4) and Eq. (5)
7     |  Compute the total loss $\mathcal{L} = \mathcal{L}_{dp} + \mathcal{L}_{sim}$ by Eq. (7) and Eq. (8)
8     |  Update current model parameters $\omega_t = \text{SGD}(\mathcal{D}^{\text{clean}}, \mathcal{L}, \omega_{t-1})$
9  **end**
   // Phase2: Adapting Model on Clean Data with FFT
10  **for** $t = 1, 2, ..., T_2$ **do**
11     |  Compute the CE loss $\ell_{ce}$ for samples in the clean subset $\mathcal{D}^{\text{clean}}$
12     |  Update current model parameters $\theta_t = \text{SGD}(\mathcal{D}^{\text{clean}}, \ell_{ce}, \theta_{t-1})$
13  **end**

---

|  | ViT-B/16 | ResNet-50 | ConvNeXt-T | MAE-ViT-B |
|---|---|---|---|---|
| Optimizer | SGD | AdamW | AdamW | AdamW |
| Learning Rate | $1\times10^{-2}$ | $1\times10^{-3}$ | $1\times10^{-4}$ | $1\times10^{-4}$ |
| Weight Decay | $1\times10^{-5}$ | $1\times10^{-5}$ | $1\times10^{-4}$ | $1\times10^{-3}$ |

Table 11: Configurations of different pre-trained models.

baselines when utilizing PEFT to adapt the pre-trained model, possibly due to the more stable training process with fewer learnable parameters. Lastly, DEFT continues to outperform all compared methods in almost all cases when applying PEFT during the model adaptation phase (DEFT w/ PEFT), and its performance can be further enhanced by replacing PEFT with FFT (DEFT w/ FFT), confirming the validity of the proposed framework.

### A.3 Additional Implementation Details

**Pseudo-code for the Proposed Framework**   DEFT is a two-phase framework consisting of a noisy label detection phase and a model adaptation phase. The pseudo-code is presented in Algorithm 1.

**Implementation of the State-of-the-arts Approaches**   To verify the effectiveness of DEFT on real-world datasets, we compare our methods with several sample selection-based state-of-the-art approaches including RoLT[40], UNICON[19], LongReMix[4], and ProMix[46]. Notably, these methods utilize CNN-like network backbones in their original implementations. To ensure a fair comparison, we implement a two-phase evaluation approach: we first run the source code to obtain the clean subset, then fully fine-tune the pre-trained CLIP model using these selected clean samples. We opt not to directly replace the model backbone because these methods incorporate various training techniques such as co-training, mixup, and contrastive learning. Removing the influence of these techniques is crucial to ensure a valid comparison.

**Hyperparameter Settings for PEFT Modules**   In the noisy label detection phase, we employ VPT [18] to adapt the visual encoder by default. In all experiments, we consistently set the length of the visual prompt to 20. For LoRA and AdaptFormer, we establish the bottleneck dimension as 8.

**Configurations of Pre-trained Models**   In the model adaptation phase, we apply FFT to a range of models that are pre-trained on ImageNet-1K. For a fair comparison, we follow the optimizer settings outlined in [1] and initialize each model with the corresponding pre-trained weights obtained from HuggingFace [1]. The specific configurations are described in Table 11.

---

[1]https://huggingface.co/

### A.4 Limitations and Broader Impacts

**Limitations**  Despite DEFT's superior performance compared to the existing methods, it exhibits certain limitations and there are several unexplored research avenues. For example, the current algorithm only deals with noisy image-label pairs in single-label multi-class classification tasks. A promising avenue for future research would be to generalize DEFT to handle noisy image-text pairs or multi-label classification settings.

**Broader Impacts**  This study falls within the domain of weakly supervised learning, a learning paradigm that aims to achieve superior performance while reducing labeling costs. Consequently, as this technique gains efficacy and wider adoption, the necessity for extensive data annotation may get diminished, potentially contributing to a rise in unemployment among data annotation professionals.

