# OpenReview forum: "Vision-Language Models are Strong Noisy Label Detectors"
_NeurIPS.cc/2024/Conference — NeurIPS 2024 poster_

### Official Review · Reviewer_mmEa · 2024-07-08

**Soundness:** 3
**Presentation:** 3
**Contribution:** 3
**Rating:** 7
**Confidence:** 5

**Summary:**

This paper proposes a novel method for learning with noisy labels leveraging pre-trained foundation models. The method is motivated by new findings that prompt learning is more robust to noisy labels when fine-tuning CLIP. The paper then designs a simple detector by learning both positive and negative prompts for each class. This paper showcases strong empirical results on multiple datasets, outperforming many existing methods.

**Strengths:**

1.	The proposed method is well-motivated. The effectiveness of CLIP is often overlooked in previous works for learning with noisy labels, and this paper examines it thoroughly and finds a good way to adapt CLIP in downstream tasks even with noisy labels.
2.	This paper makes an important contribution that prompt learning is robust to robust to noisy labels, while full fine-tuning is better on clean datasets. Based on this observation, the paper proposes a two-stage method.
3.	This paper proposes a novel idea by jointly learning positive and negative prompts for classes, and leverage the negative learning loss to optimize learnable prompts.
4.	Extensive experiments on both synthetic and real-world datasets show that the proposed method consistently outperforms existing baselines.

**Weaknesses:**

1.	It is unclear why positive and negative prompts can help detect noisy labels. It is suggested to provide some visualization examples.

**Questions:**

1.	Can examples detected as noisy label help improve the performance?

**Limitations:**

The authors have adequately addressed the limitations and potential negative societal impact of their work.

---

> ### Author Rebuttal · Authors · 2024-08-05
>
> Dear Reviewer mmEa,
>
> We sincerely appreciate the reviewer for the thoughtful feedback. We are encouraged by comments like *The proposed method is well-motivated* and *This paper makes an important contribution*. We address your concerns one by one.
>
> > W1. It is unclear why positive and negative prompts can help detect noisy labels.
>
> The primary benefit of utilizing dual textual prompts is their ability to facilitate adaptive and instance-specific selection. In contrast, the conventional Small-loss strategy presents two significant limitations:
>
> 1) The Small-loss strategy requires the selection threshold to be manually set. The efficacy of noise detection heavily depends on this threshold, and finding an optimal value typically necessitates prior knowledge, such as the noise ratio in the dataset.
> 2) The Small-loss strategy assumes that all samples with high loss values are inherently noisy. However, it overlooks the presence of hard but clean samples that also exhibit high losses. By sorting samples based solely on loss and selecting only those with lower losses as clean, the Small-loss strategy risks overlooking these challenging yet valid instances.
>
> By leveraging dual prompts, sample selection can be based on the similarity between each sample and the corresponding two prompts, i.e. $sim(I_i, T_k^+) > sim(I_i, T_k^-)$. This is equivalent to learning an adaptive threshold for each sample in a data-driven manner, thereby circumventing the limitations of the Small-loss strategy.
>
> > Q1. Can examples detected as noisy label help improve the performance?
>
> We can make use of noisy data for further improvement by treating noisy data as unlabeled data and leveraging semi-supervised techniques. Specifically, by simply incorporating FixMatch [1] in the second stage of DeFT, we improve the test accuracy on noisy CIFAR-100 with 60\% symmetric noise from 85.72\% to 86.13\%.
>
> [1] Sohn K, Berthelot D, Carlini N, et al. Fixmatch: Simplifying semi-supervised learning with consistency and confidence. NeurIPS 2020.

---

### Official Review · Reviewer_FyAK · 2024-07-08

**Soundness:** 3
**Presentation:** 3
**Contribution:** 3
**Rating:** 6
**Confidence:** 4

**Summary:**

This paper focuses on improving the finetuning performance of pretrained vision models by removing the noisy labeled data. Specifically the authors propose a two-stage method: the first stage is to learn a noise detector by prompt learning of the text encoder in CLIP. The second stage is to full-finetune the pretrained models (including CLIP and other pretrained models) . In the experiment, the authors include both synthetic datasets and real datasets and compare their model with different 6 different methods. They also use the filtered dataset to finetune different pretrain models.

**Strengths:**

1. The paper is well writen and easy to follow.
2. The analysis of the relationship between noisy data and finetune methods is very insightful (Figure 1)
3. The proposed two-stage method is reasonable and results in good performance compared to other baselines.
4. It is great to show that the filtered data works for both CLIP models and non-CLIP models even though the filtering is done by the model based on CLIP.

**Weaknesses:**

1. The optimization part is a bit unclear. (See Questions Below)
2. The proposed noisy detector cannot be reusable when working on a different dataset. Each downstream dataset needs a specific trained detector.
3. Noisy label problem is a more severe problem during the pretraining than the finetuning. But this paper only focuses on the finetuning, which makes the research scope narrow in this case.

**Questions:**

1. I am confused about the optimization for the positive prompts. Are positive prompts learnt together with negative prompts and Visual PEFT by using $L_{dp} + L_{sim}$? Or are they learned separately?
2. Can a noisy detector learned on one dataset transferred to another dataset? For example, the noisy detector learned on ImageNet can be used to detect noise in CIFAR100 dataset since there are a great number of class overlapping in these two datasets?

**Limitations:**

The authors have included the discussion of limitations in the supplementary.

---

> ### Author Rebuttal · Authors · 2024-08-05
>
> Dear Reviewer FyAK,
>
> We sincerely appreciate the reviewer for the thoughtful feedback. We are encouraged by comments like *The analysis of the relationship between noisy data and finetune methods is very insightful* and *is well-written and easy to follow*. We address your concerns one by one.
>
> > W1. The optimization part is a bit unclear.
> >
> > Q1. Are positive prompts learnt together with negative prompts and Visual PEFT by using $L_{dp} + L_{sim}$? Or are they learned separately?
>
> Sorry for the confusion. The positive prompts are **learnt together** with the negative prompts and Visual PEFT by $L_{dp} + L_{sim}$ in the first stage. We summarized the proposed method in **Algorithm 1** in the Appendix of the paper.
>
> > W2. The proposed noisy detector cannot be reusable when working on a different dataset.
> >
> > Q2. Can a noisy detector learned on one dataset transferred to another dataset?
>
> Though the main focus of this paper is to identify noisy labels in the specific downstream data, the proposed noisy label detector can also generalize to a different dataset. To validate this, we use the noisy Tiny-ImageNet dataset ($64$$\times$$64$ size) to train the noisy label detector and test it on the ImageNet dataset with 40% symmetric noise. The reason we do not use the noisy CIFAR-100 dataset for training is due to the difficulty in constructing a consistent class name mapping between CIFAR-100 and ImageNet (e.g., "turtle" in CIFAR-100 is referred to as "loggerhead" in ImageNet).
>
> The experimental results show that the F1-score on the noisy ImageNet dataset reaches 95.16%, which is 10.16% higher than the Zero-shot baseline. Besides, we find that the proposed noisy label detector can also generalize to other types of noisy labels. We construct a test dataset comprising 50% instance-dependent label noise to verify the effectiveness of the detector trained with symmetric noise and achieve an F1-score of 97.88%. These results demonstrate the strong generalization ability of the proposed noisy label detector.
>
> > W3. Noisy label problem is a more severe problem during the pretraining than the finetuning. But this paper only focuses on the finetuning, which makes the research scope narrow in this case.
>
> The noisy label problem during fine-tuning is significant because it directly impacts the model's performance on specific downstream tasks. To address this challenge, we propose the DeFT framework, which significantly improves the robustness of models against noisy labels, making a valuable contribution to the field.
>
> While our focus is on fine-tuning, we acknowledge that exploring the noisy label problem during pretraining is an intriguing avenue for future research. We hope our work provides meaningful insights and promotes further studies on this problem.

---

> > ### Comment · Reviewer_FyAK · 2024-08-12
> >
> > The authors address all my concerns and questions. Thanks for the effort in making the rebuttal. I changed my rating from 5 to 6.

---

> > > ### Author Response · Authors · 2024-08-12
> > > **Response by Authors**
> > >
> > > Dear Reviewer FyAK,
> > >
> > > Thank you for your thoughtful suggestions and the positive feedback on our work.

---

### Official Review · Reviewer_6CXy · 2024-07-10

**Soundness:** 2
**Presentation:** 3
**Contribution:** 2
**Rating:** 5
**Confidence:** 4

**Summary:**

The paper proposes a method for detecting noisy samples using vision-language models (CLIP). The main idea is to efficiently adapt (via prompt tunning) the clip model on noisy data and use this adapted model to select clean data. In the second stage, the clean data can be used to fully fine-tune a backbone model. The paper proposes the following contributions: a) using two learnable text prompts (positive and negative) to avoid using a threshold when detecting noise b) using negative learning [40] c) using a two-stage approach to adapt to the noisy distribution (first select clean samples then fully fine-tune using them). Experiments are done on multiple datasets with synthetic (CIFAR-100, Tiny-ImageNet, Stanford-Cars, CUB-200-2011) or real noisy labels (CIFAR-100N, Clothing1M, WebVision).

**Strengths:**

* S1. Good direction of efficiently adapting CLIP models for noise detection.
* S2. The general setting is sound, of using efficiently fine-tuned CLIP to select clean data and then fine-tuning using it.
* S3. Good results on multiple noisy datasets.

**Weaknesses:**

* W1. There needs to be more fair comparisons with baselines that are trained in the same settings and use similar models.

* W2. Table 1 seems to compare DEFT which has a fine-tuning stage on the noisy dataset with the initial clip model that is not trained or adapted at all. Is this the case? Is the small-loss baseline using the initial CLIP model, without any adaptation to the current dataset?

* W2.2. More appropriate baselines would be CLIP models that are adapted in standard ways to the current noisy dataset. Thus, one aspect that needs to be ablated is the fine-tuning method (none, fully fine-tuning, efficient fine-tuning, etc). Then, given a CLIP model, the second aspect that needs ablation is the selection method. Here the paper already compares against zero-shot and small-loss selection, but using the initial, pre-trained CLIP. The same should be done using an adapted clip.

* W2.3. Another simple selection method that, like DEFT doesn’t require a threshold would be to select samples with $sim(I_i, T_k) > th$ (where th=0) where $T_k$ is the text features corresponding to the correct class. This selection should be different from the zero-shot selection because the threshold is applied directly on the similarity of the image and correct class features, without applying softmax, thus without taking into account if the correct class is the class predicted by the zero-shot model.

* W3. Table 2 and Table 3 compare against multiple methods for training in noisy settings, starting with the same visual backbone, i.e. the visual part of CLIP. Is this correct? The question arises, is DEFT using additional information since it does the selection using both the visual and textual part of CLIP? Thus, the DEFT has a possible unfair advantage.

* W4. What is the motivation for using two text prompts (positive and negative)? If only one prompt is used (as usual), we can still make a prediction and optimize this prediction to be the correct class and then apply a fixed threshold (>0.5). Why shouldn’t the simple approach work and what benefits does the dual prompt bring?

**Questions:**

* Is the $L_{sim}$ loss (Eq.8) used for the adaptation in the second stage, or is it used after the selection in phase 1? If the latter, would this mean that this loss updates the PEFT module and the learnable prompts, and then the final model is used in the second stage?

**Limitations:**

The method is not sufficiently evaluated against fair baselines, see weak points for more details.

---

> ### Author Rebuttal · Authors · 2024-08-05
>
> Dear Reviewer 6CXy,
>
> We sincerely appreciate the reviewer for the thoughtful feedback. We are encouraged by comments like *The general setting is sound* and *Good results on multiple noisy datasets*. We address your concerns one by one.
>
> > W1~W2.2. There needs to be more fair comparisons with baselines in Table 1.
>
> To improve the clarity, we would like further to explain the experimental settings of the two baselines:
>
> 1) Zero-shot uses the initial CLIP model to predict labels for the training set. Samples where the given label matches the model's prediction are considered clean. In order to distinguish such an approach from model fine-tuning methods, we rename it as the **"Label-match"** strategy in the following discussion.
> 2) Small-loss selects a proportion of samples with small loss *during training* as clean samples. Therefore, the Small-loss baseline in Table 1 also utilizes the adapted CLIP model like our method, ensuring a fair comparison.
>
> To make a fair comparison between the Label-match strategy and our method, we conduct an ablation study on noisy CIFAR-100 based on the CLIP model adapted with parameter-efficient fine-tuning (PEFT). The Table below presents the F1-score for noisy label detection by three methods. It can be seen that our method outperforms the other two baselines under varying noise ratios.
>
>
> | Method                  | *sym.* 0.2 | *sym.* 0.4 | *sym.* 0.6 |
> | ----------------------- | :--------: | :--------: | :--------: |
> | **Label-match w/ PEFT** |   95.64   |   94.54   |   93.06   |
> | **Small-loss w/ PEFT**  |   97.01   |   95.08   |   91.79   |
> | **DeFT (ours) w/ PEFT** | **98.63** | **98.33** | **97.15** |
>
> Furthermore, we agree with the suggestion to perform ablations on the fine-tuning methods. Accordingly, we use a CLIP model fully fine-tuned on noisy CIFAR-100 for sample selection and present the F1-score in the table below. Comparing the results of the two tables, we observe that:
>
> 1) Regardless of the fine-tuning method used, our method (DeFT) achieves superior results.
> 2) Parameter-efficient fine-tuning (PEFT) performs better under high noise conditions, which is why we choose PEFT for adapting the CLIP model in the noisy label detection stage.
>
>
> | Method                 | *sym.* 0.2 | *sym.* 0.4 | *sym.* 0.6 |
> | ---------------------- | :--------: | :--------: | :--------: |
> | **Label-match w/ FFT** |   96.17   |   93.11   |   85.10   |
> | **Small-loss w/ FFT**  |   97.16   |   94.77   |   89.81   |
> | **DeFT (ours) w/ FFT** | **98.76** | **97.44** | **90.03** |
>
> > W2.3. Select samples with $sim(I_i, T_k) > 0$.
>
> In practice, selecting samples with $sim(I_i, T_k) > 0$ is not effective. Our experiments on CIFAR-100 with 60% symmetric noise using the initial CLIP model reveal that the minimum $sim(I_i, T_k)$ value among all samples is 0.098. As a result, setting the threshold at $sim(I_i, T_k) > 0$ does not exclude any samples and is thus not useful for filtering out noisy samples. However, when we set the threshold to 0.25, the F1-score for noise detection improved significantly to 87.57%. This indicates that the effectiveness of sample selection based on $sim(I_i, T_k)$ is highly dependent on the choice of threshold.
>
> > W3. Table 2 and Table 3 compare against multiple methods with the same visual backbone. Is DEFT using additional information since it does the selection using both the visual and textual part of CLIP? Thus, the DEFT has a possible unfair advantage.
>
> **All methods in Table 2 and Table 3 use the same backbone**, specifically the visual part of CLIP, and our method indeed leverages additional text information for sample selection. However, we do not view this as a drawback but rather one of the main contributions of this paper. To achieve better noise detection performance than previous unimodal methods, we introduce the text modality within the CLIP model. Through the DeFT framework, we successfully utilize multimodal information to enhance noise detection, achieving superior results compared to prior unimodal approaches. To the best of our knowledge, DeFT is the first method to effectively leverage multimodal information in label-noise learning. Similar explorations have also appeared in other domains, such as multi-label classification [1], semi-supervised learning [2], and out-of-distribution detection [3].
>
> > W4. What is the motivation for using two text prompts (positive and negative)? What benefits does the dual prompt bring?
>
> The key advantage of using dual textual prompts is that they enable adaptive and instance-specific selection. With a single prompt, a fixed threshold must be manually set. As discussed in the response to W2.3, finding an appropriate fixed threshold is challenging and often requires prior knowledge, such as the noise ratio. However, by using dual prompts, sample selection can be based on the similarity between each sample and the corresponding two prompts, i.e. $sim(I_i, T_k^+) > sim(I_i, T_k^-)$. This is equivalent to learning an adaptive threshold for each sample in a data-driven manner, eliminating the need for manual tuning.
>
> > Q1. Is the $L_{sim}$ loss (Eq.8) used for the adaptation in the second stage, or is it used after the selection in phase 1?
>
> Sorry for the confusion. $L_{sim}$ loss is only used in the first stage. With the filtered data, we can adapt any visual backbones using the selected clean data in the second stage. We summarized the proposed method in **Algorithm 1** in the Appendix of the paper.
>
> [1] Abdelfattah R, Guo Q, Li X, Wang X, Wang S. Cdul: Clip-driven unsupervised learning for multi-label image classification. ICCV 2023.
>
> [2] Mo S, Kim M, Lee K, Shin J. S-clip: Semi-supervised vision-language learning using few specialist captions. NeurIPS 2023.
>
> [3] Ming Y, Cai Z, Gu J, Sun Y, Li W, Li Y. Delving into out-of-distribution detection with vision-language representations. NeurIPS 2022.

---

> > ### Comment · Reviewer_6CXy · 2024-08-09
> > **Response to rebuttal**
> >
> > I thank the authors for the response. I appreciate the clarification of the Small-loss baseline and the additional ablations.
> >
> > In W2.3. you show that the initial, pre-trained CLIP model is sensible to the threshold and it makes sense. How about the adapted CLIP model, as asked in W4?
> >
> > Overall I tend to increase my score to 5.

---

> > > ### Author Response · Authors · 2024-08-10
> > >
> > > Dear Reviewer 6CXy,
> > >
> > > We sincerely appreciate your valuable comments and encouraging feedback.
> > >
> > > For the adapted CLIP model in W4, the minimum $sim(I_i, T_k)$ is $-0.443$, and the F1-scores for noise detection at different thresholds are presented in the table below. These results indicate that the adapted CLIP model is also sensitive to the choice of threshold. We will add the new baseline in the revised version.
> > >
> > >
> > > | Threshold |  0.0  |  0.1  |  0.2  |  0.3  |  0.4  |  0.5  |
> > > | --------- | :---: | :---: | :---: | :---: | :---: | :---: |
> > > | F1-score  | 80.77 | 85.72 | 89.89 | 91.93 | 92.75 | 91.81 |

---

### Official Review · Reviewer_g1PL · 2024-07-12

**Soundness:** 3
**Presentation:** 3
**Contribution:** 3
**Rating:** 5
**Confidence:** 2

**Summary:**

The paper introduces a Denoising Fine-Tuning (DEFT) framework to address the challenge of noisy labels in vision-language models, particularly focusing on models like CLIP. The DEFT framework leverages the robust alignment of textual and visual features pre-trained on extensive image-text pairs to filter out noisy labels. This is achieved by learning class-specific positive and negative textual prompts. Positive prompts highlight distinctive class features, while negative prompts act as thresholds to differentiate between clean and noisy samples. The framework uses parameter-efficient fine-tuning (PEFT) to adapt the visual encoder to align with the textual prompts. Extensive experiments on synthetic and real-world noisy datasets demonstrate that DEFT significantly improves both noisy label detection and image classification performance.

**Strengths:**

1. This paper proposes to combining textual and visual prompts for noisy label detection, which enhances the robustness of vision-language models to label noise.

2. The framework's generalizability to various pre-trained models and its parameter efficiency make it a versatile solution.

3. The experimental validation on multiple datasets, including real-world noisy data, provides strong evidence of the method's effectiveness.

4. The use of PEFT to maintain the generalization ability of pre-trained models while adapting to specific tasks is particularly noteworthy.

**Weaknesses:**

1. The heavy reliance on pre-trained models may limit the framework's applicability in scenarios where such models are not available or suitable.

2. The discussion on the practical implementation and potential limitations in different real-world settings is relatively limited.

3. The computational overhead associated with maintaining and fine-tuning dual prompts could be a concern in resource-constrained environments.

**Questions:**

1. How does DEFT perform when applied to pre-trained models with varying degrees of generalizability and domain relevance?

2. Can DEFT be adapted to scenarios with extremely high noise ratios or highly imbalanced datasets, and what modifications would be necessary to maintain its effectiveness?

3. What are the potential trade-offs between the computational overhead of DEFT and its performance gains, and how can the framework be optimized for deployment in resource-constrained environments?

**Limitations:**

The authors have discussed the limitations of this paper, and there is no negative societal impact.

---

> ### Author Rebuttal · Authors · 2024-08-05
>
> Dear Reviewer g1PL,
>
> We sincerely appreciate the reviewer for the thoughtful feedback. We are encouraged by comments like *a versatile solution* and *strong evidence of the method's effectiveness*. We address your concerns one by one.
>
> > W1. The heavy reliance on pre-trained models may limit the framework's applicability.
> >
> > Q1. How does DEFT perform when applied to pre-trained models with varying degrees of generalizability and domain relevance?
>
> **[prevalence of pre-trained models]** Pre-trained models have become a cornerstone in many tasks due to their ability to capture generalizable features from large datasets, such as multi-label classification [1], long-tailed learning [2], and out-of-distribution detection [3]. This paper focuses on noisy label detection, where the pre-trained CLIP model is available and suitable for our method in almost all scenarios.
>
> **[various pre-trained models]** In Table 4 of the paper, we compared various pre-trained models in addition to CLIP. The results validate the effectiveness of our method.
>
> **[domain relevance]** In addition, we conduct experiments on the MNIST dataset. MNIST consists of monochrome images of handwritten digits and has been verified to have a low domain relevance with CLIP as there is no data overlap with the CLIP pre-training data [4]. Results (F1-score) in the table below exhibit that our method consistently outperforms the small-loss baseline in sample selection performance.
>
> | Method          | *sym.* 0.2 | *sym.* 0.4 | *sym.* 0.6 |
> | --------------- | :--------: | :--------: | :--------: |
> | **Small-loss**  |   97.52   |   96.16   |   94.18   |
> | **DeFT (ours)** | **99.63** | **99.30** | **98.54** |
>
> > W2. The discussion on the practical implementation and potential limitations in different real-world settings is relatively limited.
>
> **[implementation]** We provided the code of this paper in the supplementary material, which contains the practical implementation details of our method on three real-world datasets. We will include more implementation details in the next version of the paper.
>
> **[limitations]** We discussed the limitations in Appendix A.4 of the paper. Specifically, our method may have some potential limitations under different real-world settings. For example, DeFT primarily focuses on the label noise problem in image classification, where the label is a class name. Therefore, it cannot directly handle the noise in image-text pair data, where the label is a text description of the image.
>
> > Q2. Can DEFT be adapted to scenarios with extremely high noise ratios or highly imbalanced datasets?
>
> We reported the noise detection results under severe noise conditions in Appendix A.1 of the Appendix. To tackle high noise ratios, there are two necessary modifications: 1) the learning rate is adjusted from $3$$\times$$10^{-2}$ to $1$$\times$$10^{-2}$, and 2) a smaller weight is assigned to the positive loss component in $L_{dp}$ to mitigate the impact of noisy pseudo-labels. For highly imbalanced datasets, we can employ class-balanced loss functions to replace the standard cross-entropy loss $\ell_{ce}$ in the model adaptation stage, such as the logit-adjustment loss [5].
>
> > W3. The computational overhead could be a concern in resource-constrained environments.
> >
> > Q3. How can the framework be optimized for deployment in resource-constrained environments
>
> It is noteworthy that we utilize parameter-efficient fine-tuning (PEFT) techniques to adapt CLIP on downstream datasets, which is both effective and efficient compared to fully fine-tuning, as PEFT is more robust to label noise and requires optimizing much fewer parameters. Even in extremely resource-constrained environments, the proposed framework can still be adjusted in the following ways:
>
> 1) **Learning transferable detector on smaller datasets**. In the first stage of DeFT, we can learn a noisy label detector on a small dataset and then transfer it to a larger one for sample selection. For example, the detector trained on Tiny-ImageNet can be used to detect noise in the overlapping classes of the ImageNet dataset.
> 2) **Adapting model with smaller backbones**. In the second stage of DeFT, the filtered data can be used with various visual backbones, as shown in Table 4 of the paper. Therefore, we can utilize a smaller model and still achieve good performance with the filtered clean data.
>
> [1] Abdelfattah R, Guo Q, Li X, Wang X, Wang S. Cdul: Clip-driven unsupervised learning for multi-label image classification. ICCV 2023.
>
> [2] Shi J, Wei T, Zhou Z, Han X, Shao J, Li Y. Long-Tail Learning with Foundation Model: Heavy Fine-Tuning Hurts. ICML 2024.
>
> [3] Ming Y, Cai Z, Gu J, Sun Y, Li W, Li Y. Delving into out-of-distribution detection with vision-language representations. NeurIPS 2022.
>
> [4] Radford A, Kim J, Hallacy C, et al. Learning transferable visual models from natural language supervision. ICML 2021
>
> [5] Menon A, Jayasumana S, Rawat A, Jain H, Veit A, Kumar S. Long-tail learning via logit adjustment. ICLR2021.

---

> > ### Author Response · Authors · 2024-08-14
> > **Official Comment by the Authors**
> >
> > Dear Reviewer g1PL,
> >
> > We appreciate your thorough evaluation and helpful suggestions and comments. In our response, we have provided point-by-point responses to your specific comments. We hope our response addresses all the concerns raised in your review.
> >
> > Since the author-reviewer discussion ends soon, we are happy to hear your thoughts if you need additional clarifications from the authors. Thank you very much.
> >
> > Best

---

### Decision · Program_Chairs · 2024-09-25

**Decision:**

Accept (poster)

**Comment:**

The paper introduces DeFT, a Denoising Fine-Tuning framework that effectively adapts vision-language models to handle noisy labeled data. The key innovation of this work lies in the use of positive and negative textual prompts to enhance the alignment of visual and textual features, which helps in detecting and mitigating the impact of noisy labels. DeFT's versatility in being applicable to various pre-trained models, along with its parameter-efficient approach, is well-supported by strong experimental results on both synthetic and real-world noisy datasets. All reviewers agree that the paper provides significant contributions and is ready for publication. The AC agrees with this assessment.